# Challenges and Opportunities for the Clinical Application of the Combination of Immune-Checkpoint Inhibitors and Radiation Therapy in the Treatment of Advanced Pancreatic Cancer

**DOI:** 10.3390/cancers17040606

**Published:** 2025-02-11

**Authors:** Masashi Kanai

**Affiliations:** Department of Clinical Oncology, Kansai Medical University Hospital, 3-1 Shinmachi 2 Chome, Hirakata City 573-1191, Osaka, Japan; kanaim@hirakata.kmu.ac.jp

**Keywords:** pancreatic ductal adenocarcinoma, immune-checkpoint inhibitors, radiation therapy, tumor microenvironment

## Abstract

Pancreatic cancer remains one of the deadliest malignancies, with slow progress in treatment. Immune-checkpoint inhibitors (ICIs) have transformed the treatment of several cancers but have shown limited efficacy in pancreatic ductal adenocarcinoma (PDAC) due to its immune-suppressive tumor microenvironment (TME). Recent clinical trials have investigated combining ICIs with radiation therapy (RT), which can potentially enhance immune responses by increasing tumor immunogenicity. This review discusses these recent efforts, highlighting their safety, efficacy, and implications for future research.

## 1. Introduction

Pancreatic ductal adenocarcinoma (PDAC) is a highly aggressive malignancy, ranking as the fourth leading cause of cancer-related deaths in the United States, and the five-year survival rate remains below 15% [1]. Immune-checkpoint inhibitors (ICIs), such as anti-PD-1/PD-L1 and anti-CTLA-4 antibodies, have revolutionized cancer treatment, demonstrating durable responses in high immunogenicity tumors such as melanoma and non-small-cell lung cancer, and their clinical application is expanding year by year [2]. However, their efficacy in PDAC has been limited due to low antigenicity to trigger an immune response and abundant cancer-associated fibroblasts, both of which result in an immune-suppressive tumor microenvironment (TME) with few tumor-infiltrating lymphocytes (TILs) [3,4].

Radiation therapy (RT) has shown immunomodulatory effects, including the upregulation of tumor-associated antigens through immunogenic cell death, leading to the recruitment of TILs in preclinical models [5,6,7]. These preclinical findings provide a rationale for combining ICIs with RT to enhance antitumor efficacy [8]; this rationale has already been tested in phase III trials in other cancers [9,10,11,12,13,14], and positive results have been reported in non-small-cell lung cancer and cervical cancer [9,10,11]. 

This review discusses recent findings on the combination of ICIs with RT and implications for future therapeutic strategies in patients with advanced PDAC.

## 2. Preclinical Data on the Combination of ICIs and RT

### 2.1. Immunomodulatory Effects of RT in Preclinical Models

Zhang et al. demonstrated that treating a xenografted fibrosarcoma tumor with local RT of 10 Gy in one fraction causes the sufficient release of tumor-associated antigens to sensitize tumor stromal cells for destruction by adoptive T-cell transfer [5]. The transient appearance of tumor-associated peptide–MHC complexes was visualized directly using high-affinity T-cell receptor tetramers. These data support the idea that RT induces immunogenic cell death and the subsequent release of tumor-associated antigens, leading to more robust T-cell activation and the eradication of tumor cells.

Deng et al. explored the effect of RT on PD-L1 expression in preclinical breast and colon cancer models [15]. Tumors in a mouse model were subjected to local RT with a single dose of 12 Gy. Three days post-RT, the tumors were harvested, and PD-L1 expression was measured using flow cytometry on various cell populations, including tumor cells, dendritic cells (DCs), myeloid-derived suppressor cells (MDSCs), and macrophages. A significant increase in PD-L1 expression in tumor cells and DCs was observed following RT. Macrophages exhibited a modest elevation in PD-L1 levels, while MDSCs maintained high baseline PD-L1 expression that remained unchanged post-RT.

Klug et al. utilized both a genetically engineered RIP1-Tag5 mouse model mimicking pancreatic insulinoma and a xenografted melanoma tumor to evaluate the effects of low-dose RT on tumor-associated macrophages (TAMs) [16]. Local RT was administered at a dose of 2 Gy to the tumor sites, and they subsequently analyzed the macrophage populations within the tumors. They found that RT induced a phenotypic shift in TAMs from an M2-like (pro-tumor) state to an M1-like (antitumor) state, characterized by increased expression of inducible nitric oxide synthase (iNOS). This M1 polarization was associated with the enhanced recruitment of TILs and improved tumor control.

Deng et al. reported that DCs act as key mediators of the innate immune response following RT [17]. After RT, damaged tumor cells released cytosolic DNA into the TME. DCs within the irradiated tumor site sensed this DNA via the cyclic GMP–AMP synthase (cGAS)–stimulator of interferon genes (STING) pathway. This cGAS-STING pathway activation resulted in the robust production of interferon-beta (IFN-β) by DCs. The increased production of IFN-β was crucial for initiating an effective antitumor immune response. IFN-β produced by DCs enhanced their capacity to prime and activate tumor-specific cytotoxic T lymphocytes. In knockout mouse models lacking components of the cGAS-STING pathway, DCs failed to produce IFN-β in response to radiation, leading to a diminished T-cell-mediated antitumor response.

### 2.2. Efficacy of Combining ICIs with RT in Preclinical Models

Dovedi et al. investigated the efficacy of combining anti-PD-L1 antibodies (mAbs) with RT using syngeneic mouse models of melanoma, colorectal, and breast cancers [18]. The mice were divided into four groups: control, RT alone, anti-PD-L1 mAb alone, and the combination of anti-PD-L1 mAb and RT. RT was delivered at 2 Gy per fraction over 10 sessions. The local tumor control, survival, and generation of tumor-specific memory immune responses were measured. The study found that RT upregulated PD-L1 expression in tumor cells through IFN-γ produced by CD8^+^ T cells. The combination of anti-PD-L1 mAb and RT demonstrated significantly enhanced tumor control and prolonged survival compared to either monotherapy. Furthermore, the combination treatment led to the development of robust tumor-specific memory immune responses, as evidenced by the ability of the treated mice to reject tumor rechallenge.

The same research group investigated whether RT could increase the diversity of the T-cell population at the irradiated tumor site [19]. They reported that the T-cell receptor (TCR) repertoire remained dominated by the polyclonal expansion of pre-existing T-cell clones, indicating that RT did not alter the diversity of the T-cell population. In addition, they evaluated both local and systemic immune responses using a dual-tumor model, where only one tumor received RT and the other did not. While RT increased TILs in the irradiated tumor, it did not in the non-irradiated tumor, suggesting that RT does not necessarily induce systemic antitumor immunity.

Fujiwara et al. investigated the efficacy of combining ICI with RT using a syngeneic orthotopic mouse model of pancreatic cancer [20]. In their experiment, pancreatic cancer cells of the KPC cell line were first grown in the subcutaneous area of the mouse, and then the grown tumors were implanted into the pancreas of another syngeneic mouse. The pancreatic tumors were irradiated with 8 Gy × three fractions between days 6 and 8 after implantation. Anti-PD-1 mAb was administered starting on day 6 and repeated every 4 days for a total of six doses. The combination of anti-PD-1 mAb and RT significantly suppressed tumor growth and prolonged overall survival compared with either monotherapy. The addition of an indoleamine 2,3-dioxygenase 1 (IDO1) inhibitor to the combination of anti-PD-1 mAb and RT did not improve the antitumor efficacy. Blood and tissue samples were collected to evaluate the immunomodulatory effects of the combination of ICI and RT. Several biomarkers involved in immune activation (e.g., IFN-γ, Cd28, and Lag 3) were significantly upregulated after the combination of ICI and RT.

In summary, these preclinical data highlight the potential of combining ICIs with RT to enhance systemic antitumor responses (Figure 1). 

## 3. Clinical Trial Data on the Safety and Efficacy of the Combination Therapy of ICIs and RT in Advanced PDAC

### 3.1. Combination of Pembrolizumab Plus Trametinib and Stereotactic Body Radiation Therapy in Patients with Locally Recurrent PDAC

The efficacy of stereotactic body radiation therapy (SBRT) combined with pembrolizumab and an MEK inhibitor, trametinib, in patients with locally recurrent PDAC was investigated in a randomized phase II trial (NCT02704156) [21]. Patients were randomized into two arms: the experimental arm (*n* = 85) received SBRT (40 Gy in five fractions) plus pembrolizumab (200 mg intravenously every 3 weeks) and trametinib (2 mg orally daily), while the control arm (*n* = 85) received SBRT plus gemcitabine (1000 mg/m^2^ intravenously on days 1 and 8 of a 21-day cycle). The primary endpoint was progression-free survival (PFS), while secondary endpoints included overall survival (OS), the objective response rate (ORR), safety, and quality of life (QOL).

The results showed that the experimental arm achieved a significantly longer median PFS compared to the control arm (8.2 versus 5.4 months, *p* < 0.001). The hazard ratio (HR) was 0.60 (95% CI: 0.44 to 0.81), which exceeded the pre-specified threshold of 0.77. The median OS was also significantly improved in the experimental arm, reaching 15.2 months versus 11.3 months in the control arm, and the HR was 0.69 (95% CI: 0.51 to 0.95; *p* = 0.021). The ORR was higher in the experimental arm (31%) compared to the control arm (17%). Grade 3 or higher AEs were observed in 31% of patients in the experimental arm and 20% in the control arm (Table 1). Changes in the QOL score were not clinically significant in either arm.

### 3.2. Combination of Durvalumab Plus Tremelimumab and SBRT in Patients with Refractory Metastatic PDAC

The safety and preliminary efficacy of combining an anti-PD-L1 antibody, durvalumab, and an anti-CTLA4 antibody, tremelimumab, with SBRT were evaluated in a phase I trial (NCT02311361) [22]. The trial aimed to assess whether dual ICIs could enhance antitumor immunity when combined with SBRT. A total of 59 patients were assigned to one of four cohorts (A1, A2, B1, or B2) according to the radiation dose or single/dual ICIs (Figure 2). SBRT was delivered at a dose of 8 Gy in one fraction on day 1 (cohorts A1 and B1) or 25 Gy in five fractions on days −3 to −1 (cohorts A2 and B2). Durvalumab was administered at 1500 mg on day 1 and every four weeks until unacceptable toxicity or disease progression occurred (cohorts A1 and A2). In addition to durvalumab, patients in cohort B received tremelimumab at a single dose of 75 mg on day 1. The trial’s primary endpoint was safety, and the secondary endpoints included PFS, OS, and immune-related changes in the TME. 

The combination of ICIs and SBRT was well tolerated in each cohort, and no unexpected AEs were observed. Grade 3–4 AEs occurred in 31.0% of 58 evaluable patients from all cohorts, and 2 patients in cohort B2 developed ICI-related colitis. No dose-limiting toxicities were observed, indicating an acceptable safety profile of the combination therapy. Among 39 patients evaluable for efficacy, 2 patients achieved a partial response, and the median PFS and OS in cohort B2 were 2.3 months (95% CI: 1.9 to 3.4 months) and 4.2 months (95% CI: 2.9 to 9.3 months), respectively, which showed no marked differences compared to the historical data (Table 1). Preliminary immune profile analysis using paired tissue samples revealed increased the CD3^+^ and CD8^+^ T cells in the TME post-treatment in four patients. 

### 3.3. Combination of Nivolumab, with or Without Ipilimumab, with SBRT in Patients with Refractory Metastatic PDAC

The efficacy and safety of combining SBRT with an anti-PD-1 antibody, nivolumab, with or without an anti-CTLA4 antibody, ipilimumab, was evaluated in a randomized phase II study (CheckPAC trial) (NCT02866383) [23]. The trial enrolled patients with metastatic PDAC that progressed beyond first-line chemotherapy. Patients in Arm A (*n* = 41) received SBRT (15 Gy in one fraction on day 1) followed by nivolumab (3 mg/kg every 2 weeks), while those in Arm B (*n* = 43) received SBRT (15 Gy in one fraction on day 1) followed by nivolumab (3 mg/kg every 2 weeks) and ipilimumab (1 mg/kg every 6 weeks). ICIs were continued until unacceptable toxicity or disease progression occurred. The primary endpoint was the clinical benefit rate, defined as the proportion of patients achieving a complete response, partial response, or stable disease lasting at least eight weeks. Secondary endpoints included safety, the ORR, PFS, and duration of response.

The clinical benefit rate was 17.1% in Arm A and 37.2% in Arm B, and Arm B met the pre-specified threshold of >30% clinical benefit rate. The ORR was 2.4% in Arm A and 14.0% in Arm B, and one patient in Arm A and six patients in Arm B showed a partial response, respectively. All seven responders were confirmed to have mismatch repair (MMR)-proficient tumors. Notably, one patient in Arm B showed a durable response and is alive at 55 months after enrollment. The median PFS was 1.7 months in Arm A and 1.6 months in Arm B. The median OS was 3.8 months in both arms. The combination therapies were generally well tolerated. Grade 3–4 treatment-related AEs occurred in 24.4% of patients in Arm A and 30.2% in Arm B, with common AEs including fatigue, diarrhea, and rash. In summary, the addition of ipilimumab to SBRT and nivolumab resulted in a higher clinical benefit rate and ORR compared to SBRT with nivolumab, although PFS and OS were similar between the two arms (Table 1). 

### 3.4. Combination of Nivolumab with CRT in Patients with Locally Advanced PDAC

We have launched a multicenter, randomized phase III trial to evaluate the efficacy of S-1-based CRT with or without nivolumab in patients with unresectable, locally advanced or borderline resectable PDAC (JCOG1908E) in 2020 (jRCT2080225361, Figure 3) [24]. The trial aims to clarify whether adding nivolumab to the standard CRT regimen improves OS in chemo-naïve patients with locally advanced PDAC. The trial targets enrollment of 210 patients across 14 institutions in Japan. The eligible participants must have histologically confirmed unresectable, locally advanced or borderline resectable PDAC and meet strict performance and organ function criteria. The patients are randomized in a 1:1 ratio to receive either S-1-based CRT alone or S-1-based CRT combined with nivolumab (Table 1).

In both arms, CRT consists of 50.4 Gy of radiation delivered in 28 fractions with concurrent oral S-1 chemotherapy (80 mg/m^2^/body) administered twice daily on days of irradiation. In the experimental arm, nivolumab is administered at a dose of 240 mg intravenously every two weeks during CRT and continued as maintenance therapy for up to one year unless the patients experience unacceptable toxicities or disease progression. Secondary endpoints include PFS, the ORR, R0 resection rate, and safety. 

The results of this trial, if successful, could establish a new standard of care for patients with locally advanced PDAC. The enrollment of 210 patients was completed in November 2024, and the topline results will be reported in 2028.

## 4. Discussion

Based on the promising preclinical data showing that RT can modulate the immune-suppressive TME to the immune-active one, three clinical trials evaluating the safety and efficacy of combining ICIs with RT in advanced PDAC have been reported [21,22,23]. All trials reported the acceptable safety profile of combining ICI with RT; however, the efficacy results were mixed. Because the disease status, radiation dose and schedule, and ICI regimen differed among the trials, we should be cautious in interpreting these data. Two clinical trials targeted the patients with refractory metastatic disease [22,23]. In Xie`s phase I trial, preliminary efficacy was evaluated, and a partial response lasting less than 2 months was observed in one patient in cohort A1 (*n* = 14), while a partial response lasting over 16.5 months was observed in one patient in cohort B2 (*n* = 16). In each cohort (A1, A2, B1, and B2), the median PFS and OS showed no marked differences compared to the historical data. In Chen’s phase II trial, the primary endpoint was set as the clinical benefit rate, and it was met in Arm B, which tested the combination of dual ICIs (nivolumab and ipilimumab) and SBRT with a single fraction of 15 Gy. One patient in Arm B showed an exceptional response and survived for more than 55 months after the study enrollment. For the patient with an MMR-deficient tumor, ICIs can activate the immune response in a tumor agnostic manner, including PDAC [25,26]. However, this exceptional responder had an MMR-proficient tumor. Therefore, the exploration of biomarkers other than MMR deficiency that can predict the potential response to the combination of ICI and RT is warranted. In contrast, the results of OS and PFS in Arm B were unsatisfactory at 3.8 and 1.6 months, respectively. There are several possible reasons for the disappointing OS and PFS results in both trials. First, patients with refractory metastatic PDAC often have a poor baseline general condition due to the cumulative damages of advanced disease, including cachexia and malnutrition [27,28]. Second, persistent inflammation driven by tumor progression leads to the accumulation of immunosuppressive cells in the TME, which hinders the antitumor immune responses of ICIs [29]. Third, patients with refractory metastatic PDAC often present with lymphopenia, likely due to prior chemotherapy or overall disease burden. A reduced pool of functional T cells significantly diminishes the antitumor immune response [30,31]. These factors undermine the ability of the immune system to mount a robust antitumor response, as a competent immune system is critical for the efficacy of ICIs [32,33].

In contrast to the unsatisfactory efficacy observed in refractory metastatic PDAC, Zhu et al. reported the positive results from a phase II trial, demonstrating an HR of 0.60 for PFS and 0.69 for OS in patients with locally recurrent PDAC after curative resection [21]. The patients in the experimental treatment arm received 25 Gy of SBRT in one fraction to the locally recurrent disease on day 1 and an MEK inhibitor, trametinib, was added to pembrolizumab. The trial targeted patients with locally recurrent PDAC who were chemo-naïve for their recurrent disease. Therefore, they were expected to have a better baseline general condition and smaller tumor volumes compared to patients with refractory metastatic disease. In fact, the baseline CA 19-9 levels were below 200 U/mL in more than 40% of patients in this trial, while the median CA 19-9 level was 3230 U/mL in Chen`s trial [23]. Similarly, the median CA 19-9 level was 5880 U/mL in cohort B2 of Xie’s trial [22]. Since patients with better general conditions and smaller tumor volumes tend to experience more favorable clinical outcomes when treated with ICIs [34], these favorable baseline patient characteristics may have influenced the positive results in Zhu’s trial. In addition, the proportion of patients with MMR deficiency in Zhu’s trial was higher (8%) compared to the low prevalence (1~2%) reported in previous studies [35,36], which may also have contributed to the positive results. On the other hand, the backbone chemotherapy differed between the two arms, with trametinib in the experimental arm and gemcitabine in the control arm. As minimal data were available on the combination of trametinib with ICIs or RT, it is difficult to estimate the contribution of trametinib to the positive results in this trial.

The combination of ICIs and RT has been tested in PDAC in a neoadjuvant setting (NCT02305186) [37]. In the phase Ib/II trial by Katz et al., 37 patients with resectable or borderline resectable PDAC were randomized into two arms: one receiving CRT alone (control arm, *n* = 13) and the other receiving CRT with pembrolizumab (combination arm, *n* = 24). CRT included 50.4 Gy of radiation delivered in 28 fractions with concurrent capecitabine (825 mg/m^2^ orally two times a day). Pembrolizumab was administered at 200 mg every three weeks on days 1, 22, and 43. The study’s primary endpoint was safety, while the second primary endpoint was the relative density of CD8+ TILs in the resected tumor specimens. OS, PFS, and the R0 resection rates were also evaluated. The patient baseline characteristics was favorable compared to those of patients with refractory metastatic disease, and the median CA 19-9 level was 108 U/m. Among 13 patients assigned to the control arm, 6 patients (46%) could not proceed to pancreatectomy due to disease progression during neoadjuvant therapy, while 7 out of 24 patients (29%) in the combination arm could not proceed to pancreatectomy due to development of distant metastasis, but none of the 7 cases experienced local disease progression. Among the patients who underwent pancreatectomy, the R0 resection rates were numerically higher in the combination arm (82%) compared to the control arm (57%). R0 resection is known to be the most important factor in determining the clinical outcome of patients with PDAC [38,39]. In terms of other efficacy data, the combination arm achieved a median OS of 27.8 months (95% CI: 17.1 to not reached [NR]) compared to 24.3 months (95% CI: 12.6 to NR) in the control arm. The median PFS was 18.2 months (95% CI: 9.4 to 27.0) in the combination arm and 14.1 months (95% CI: 2.6 to 24.3) in the control arm. The R0 resection rates were 82% and 57% in the combination and the control arms, respectively. Given the limited sample size (*n* = 37), further studies in patients with favorable general conditions and localized tumor are warranted. Post-treatment immune profile analysis using resected tumor tissues showed numerically higher CD8^+^ TILs in the combination arm (67.4 cells/mm^2^, *n* = 17) compared to the control arm (37.9 cells/mm^2^, *n* = 7). Numerically better trends were observed in the density of other T-cell populations in the combination arm; however, the authors concluded that the combination therapy failed to enhance the immune response due to the lack of a significant difference between the two arms.

We have launched a multicenter, randomized phase III trial (JCOG1908E) in 2020 to clarify whether the combination of nivolumab with S-1-based CRT could improve OS in patients with unresectable, locally advanced or borderline resectable PDAC [24]. In this trial, we are targeting the chemo-naïve patients with locally advanced PDAC. These criteria could help to select the patients whose immune systems are not severely compromised due to prior chemotherapy or a high tumor burden. The trial protocol includes other rigorous patient selection/exclusion criteria and treatment schedules, all of which are critical to obtaining robust results on the efficacy of combining ICIs with CRT. If successful, this approach could be established as a standard treatment option for patients with locally advanced PDAC. The enrollment of 210 patients was completed in November 2024, and the topline results will be reported in 2028.

Innovative strategies are warranted to take full advantage of the combination of ICI and RT in PDAC. The integration of novel therapeutic modalities, such as toll-like receptor 9 agonist (SD-101, NCT04050085) or toll-like receptor 7/8 agonist (BDB001, NCT03915678), have been tested in patients with refractory metastatic PDAC [40,41]. In addition, personalized vaccines [42] or pan-KRAS inhibitors [43], may offer new avenues for improving the efficacy of the ICI and RT combination in the future. Research on biomarker-driven approaches, such as stratifying patients based on the immune profile of the TME, is also relevant to identify subsets of patients who are more likely to respond to ICI-based therapy. While challenges persist, continuing efforts in this area will provide valuable insights and facilitate the development of ICI-based therapies for PDAC.

## 5. Conclusions

Combining ICIs with RT in PDAC is a promising but challenging approach. While recent clinical trials have demonstrated safety, their clinical efficacy has been limited, particularly in refractory metastatic disease. To maximize the benefit of ICI-based therapy in this devastating disease, it may be important to administer ICIs before the deterioration of general conditions, including the immune system. The ongoing JCOG1908E trial will clarify whether the combination of ICI and CRT could improve OS in chemo-naïve patients with unresectable, locally advanced or borderline resectable PDAC in the future.

## Figures and Tables

**Figure 1 cancers-17-00606-f001:**
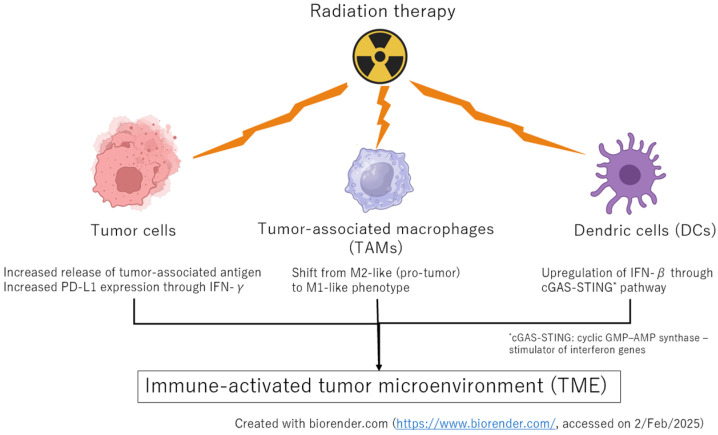
RT can influence tumor cells, TAMs, and DCs and modulate the TME.

**Figure 2 cancers-17-00606-f002:**
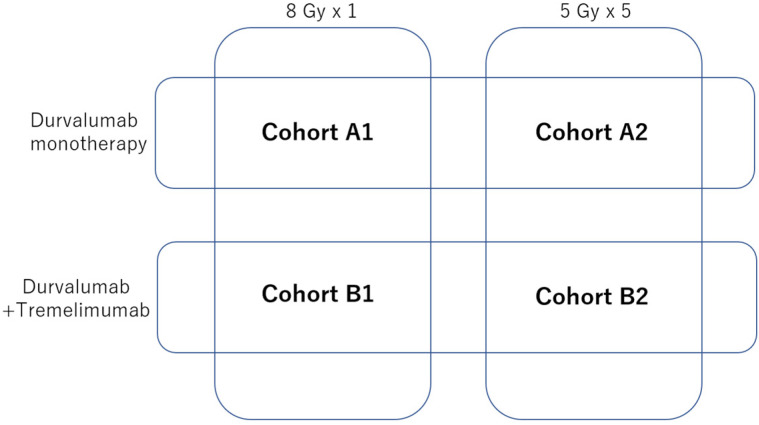
Radiation dose and ICI regimen of each cohort.

**Figure 3 cancers-17-00606-f003:**
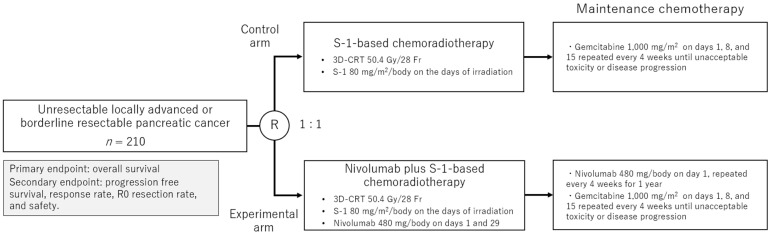
Study schema of JCOG1908E trial.

**Table 1 cancers-17-00606-t001:** Pivotal clinical trials testing the combination therapy of ICI and RT in advanced PDAC.

	Zhu et al. [21]	Xie et al. [22]	Chen et al. [23]	Sano et al. [24]
Study design	Phase II	Phase I	Phase II	Phase III
Sample size	170	59	84	216
Median age (range)	65 (experimental arm) (54–74)	62 (cohort A1) (43–80)	63 (Arm A) (35–80)	Not available
Gender (male/female)	105/65	37/22	44/40	Not available
Disease status	Local recurrence after curative resection	Metastatic	Metastatic	Locally advanced Borderline resectable
Treatment line	First line	Second line or later	Second line or later	First line
Stage	Not applicable	IV	IV	I/II/III
RT	SBRT 40 Gy/5 fractions	SBRT 5 Gy/one fraction (cohorts A1 and B1) or 25 Gy/5 fractions (cohorts A2 and B2)	SBRT 15 Gy/one fraction	S-1 based RT 50.4 Gy/28 fractions
ICI regimen	No ICI + gemcitabine (control arm) Pembrolizumab + MEK inhibitor (experimental arm)	Durvalumab monotherapy (cohorts A1 and A2) Durvalumab+ Tremelimumab (cohorts B1 and B2)	Nivolumab monotherapy (Arm A) Nivolumab +Ipilimumab (Arm B)	No ICI (control arm) Nivolumab (experimental arm)
Grade 3–4 AEs	20% (control arm) 31% (experimental arm)	7.1% (cohort A1) 33.3% (cohort A2) 21.1% (cohort B1) 62.5% (cohort B2)	24% in Arm A 31% in Arm B	Ongoing
Efficacy	PFS and OS showed significant improvements in the experimental arm	PFS and OS showed no marked improvement compared to the historical data	The clinical benefit rate exceeded the pre-specified threshold in Arm B	Ongoing

## Data Availability

No new data were created, and all data are publicly available.

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
