# Peer review of "Challenges and Opportunities for the Clinical Application of the Combination of Immune-Checkpoint Inhibitors and Radiation Therapy in the Treatment of Advanced Pancreatic Cancer"

_cancers, 2025, doi:10.3390/cancers17040606_

Round 1
Reviewer 1 Report
Comments and Suggestions for Authors
The review by Masashi Kanai focuses on the combination of immune checkpoint inhibitors and radiotherapy for the treatment of pancreatic cancer patients.
The review is well written and I think the work is interesting, but there are some points that need to be clarified to have a general overview of the possibility of combining different therapeutic strategies for the treatment of this aggressive tumor.
Line 32: Various papers have suggested an increase in the survival of PDAC patients. Today, the survival of PDAC patients is around 12-13%.
Line 49: specify the tumor type
Line 58: specify which preclinical tumor model
Line 66: specify the tumor type
Figure 1: improve the graphical aspect of the figure, for example by using cell drawings before and after radiotherapy
Line 112: introduce the acronym CRT
Lines 114-115: add clinical trial number and sponsor
Line 115: add the target of pembrolizumab
Lines 144-146: add clinical trial number and sponsor
Line 145: add the target of trametinib
Line 164: add the target of tremelimumab
Lines 163-165: add clinical trial number and sponsor
Line 167: the cohort of patients was already present in line 165, please remodulate the sentence
Line 188: add the target of nivolumab
Lines 187-189: add the number of Clinical trial
-To give a complete overview of ongoing trials combining ICIs and RT, the author can include the clinical trials NCT04050085 and NCT03915678.
-Among the studies on mouse models, the paper by Fujiwara K et al PMID 32675194 may be of interest.
-In the conclusion section, suggest other therapies that can be added to RT and immune checkpoint inhibitors to improve survival, especially in patients with advanced and metastatic disease.
Author Response
I appreciate the reviewers for their thoughtful comments about the manuscript.
I have carefully considered the comments and have revised the manuscript according to the reviewers’ comments. The revised portions of the manuscript are in red.
Reviewer 1
The review by Masashi Kanai focuses on the combination of immune checkpoint inhibitors and radiotherapy for the treatment of pancreatic cancer patients.
The review is well written and I think the work is interesting, but there are some points that need to be clarified to have a general overview of the possibility of combining different therapeutic strategies for the treatment of this aggressive tumor.
Line 32: Various papers have suggested an increase in the survival of PDAC patients. Today, the survival of PDAC patients is around 12-13%.
Reply: The sentence is revised as follows.
“the five-year survival rate remains below 15%” (Line 32)
Line 49: specify the tumor type
Reply: Tumor type is fibrosarcoma and this information is added to the revised manuscript.
Line 58: specify which preclinical tumor model
Reply: The preclinical tumor model includes breast and colon cancer and this information is added to the revised manuscript.
Line 66: specify the tumor type
Reply: Tumor is derived from pancreatic beta-cells and this information is added to the revised manuscript.
Figure 1: improve the graphical aspect of the figure, for example by using cell drawings before and after radiotherapy
Reply: I have recreated the figure 1 using the BioRender.com.
Line 112: introduce the acronym CRT
Reply: The acronym of the CRT is added to the revised manuscript.
Lines 114-115: add clinical trial number and sponsor
Reply: The clinical trial number is added to the revised manuscript. Since the clinical trial number is considered to be sufficient to identify the trial, sponsor information is not added.
Line 115: add the target of pembrolizumab
Reply: The target of pembrolizumab is added to the revised manuscript.
Lines 144-146: add clinical trial number and sponsor
Reply: The clinical trial number is added to the revised manuscript.
Line 145: add the target of trametinib
Reply: The target of trametinib is added to the revised manuscript.
Line 164: add the target of tremelimumab
Reply: The target of tremelimumab is added to the revised manuscript.
Lines 163-165: add clinical trial number and sponsor
Reply: The clinical trial number is added to the revised manuscript.
Line 167: the cohort of patients was already present in line 165, please remodulate the sentence
Reply: Sentence was revised according to the reviewer’s suggestion.
Line 188: add the target of nivolumab
Reply: The target of nivolumab is added to the revised manuscript.
Lines 187-189: add the number of Clinical trial
Reply: The clinical trial number is added to the revised manuscript.
-To give a complete overview of ongoing trials combining ICIs and RT, the author can include the clinical trials NCT04050085 and NCT03915678.
Reply: Thank you for sharing the valuable information. NCT04050085 and NCT03915678 trials are included in the discussion section in the revised manuscript.
-Among the studies on mouse models, the paper by Fujiwara K et al PMID 32675194 may be of interest.
Reply: Thank you for sharing the valuable information. The paper by Fujiwara K et al. is included in the section 2 “Preclinical data on the combination of ICIs and RT” of the revised manuscript.
-In the conclusion section, suggest other therapies that can be added to RT and immune checkpoint inhibitors to improve survival, especially in patients with advanced and metastatic disease.
Reply: In the conclusion section, following text is added.
“To maximize the benefit of ICI-based therapy in this devastating disease, it may be im-portant to administer ICIs before the deterioration of general conditions including im-mune systems.”
Reviewer 2 Report
Comments and Suggestions for Authors
Line 30-31:“Pancreatic ductal adenocarcinoma (PDAC) is a highly aggressive malignancy, ranking as the fourth leading cause of cancer-related deaths worldwide”
Please show the source.
Is it [1]?
Line41: [5, 6, 7]→[5-7]
Line43: [9, 10, 11, 12, 13, 14]→[9-14]
Line44: [9, 10, 11]→[9-11]
Line 58, 74: Deng et al. → Deng et al.
Line 66: Klug et al. → Klug et al.
Line 87: Dovedi et al. → Dovedi et al.
Line 93, 121, 132: CD8+ T cells. → CD8+ T cells.
Figure 1. There is a red wavy line under the cGAS.
It would be easier to also explain the abbreviations of Figure 1 in the footnotes of Figure 1.
Line 150: “cycle” or “cycles”
Line 163: Xie et al. → Xie et al.
Line 184: CD8 positive → CD8+ ??
Line 187: Chen et al. → Chen et al.
In table 1: et al → et al
It may be easier to indicate disease stages by No. (I to VI).
Age and gender information, if available, is also easy to understand.
Font sizes must be decreased. 9-10pt
Line233: [22 23, 24, 25]→[22-25]
Line 258: Zhu et al. → Zhu et al.
Line 278: Katz et al. → Katz et al.
What are the expectations of the combination therapy and what can be attributed to the results of each report?
Since this article is a review paper, it is useful to illustrate the author's ideas or hypotheses about the mechanisms commonly found in each reference paper for easy understanding for readers.
The impression is that there are many lines breaks throughout. I think it would be good to summarize where it can be summarized.
Author Response
I appreciate the reviewers for their thoughtful comments about the manuscript.
I have carefully considered the comments and have revised the manuscript according to the reviewers’ comments. The revised portions of the manuscript are in red.
Reviewer 2
Line 30-31:“Pancreatic ductal adenocarcinoma (PDAC) is a highly aggressive malignancy, ranking as the fourth leading cause of cancer-related deaths worldwide”
Please show the source.
Is it [1]?
Reply: I have added the reference number and revised the sentence as follows.
Pancreatic ductal adenocarcinoma (PDAC) is a highly aggressive malignancy, rank-ing as the fourth leading cause of cancer-related deaths in the United States [1]
Line41: [5, 6, 7]→[5-7]
Line43: [9, 10, 11, 12, 13, 14]→[9-14]
Line44: [9, 10, 11]→[9-11]
Line 58, 74: Deng et al. → Deng et al.
Line 66: Klug et al. → Klug et al.
Line 87: Dovedi et al. → Dovedi et al.
Line 93, 121, 132: CD8+ T cells. → CD8+ T cells.
Reply: I have revised the manuscript according to the above mentioned reviewer’s suggestions.
Figure 1. There is a red wavy line under the cGAS.
Reply: I have deleted the red wavy lines.
It would be easier to also explain the abbreviations of Figure 1 in the footnotes of Figure 1.
Reply: Figure 1 abbreviations are spelled out in the revised Figure..
Line 150: “cycle” or “cycles”
Line 163: Xie et al. → Xie et al.
Line 184: CD8 positive → CD8+ ??
Line 187: Chen et al. → Chen et al.
Reply: I have revised the manuscript according to the above mentioned reviewer’s suggestions.
In table 1: et al → et al
It may be easier to indicate disease stages by No. (I to VI).
Age and gender information, if available, is also easy to understand.
Font sizes must be decreased. 9-10pt
Reply: I have added the information of the disease status, the age, the gender, and the treatment line in the revised Table 1. In addition, font size has been reduced to 9pt..
Line233: [22 23, 24, 25]→[22-25]
Line 258: Zhu et al. → Zhu et al.
Line 278: Katz et al. → Katz et al.
Reply: I have revised the manuscript according to the above mentioned reviewer’s suggestions.
What are the expectations of the combination therapy and what can be attributed to the results of each report?
Since this article is a review paper, it is useful to illustrate the author's ideas or hypotheses about the mechanisms commonly found in each reference paper for easy understanding for readers.
Reply: I have revised the Figure 1 so that readers can easily see the points for the rationale of combining RT with ICI. In addition, the study shema of ongoing JCOG1908E trial, which we have designed, is includedas Figure 2.
The impression is that there are many lines breaks throughout. I think it would be good to summarize where it can be summarized.
Reply: I have minimized line breaks throughout the revised manuscript.
Reviewer 3 Report
Comments and Suggestions for Authors
This is an informative review paper on a topic that is interesting to clinicians and basic scientists alike.
Major aspects:
1. Targeting resectable PDAC in a neoadjuvant setting requires thourough definition of the terms resectable, borderline, advanced. The trial mentioned (Katz et al.) has to be put into context of ongoing trials on neoadjuvant chemotherapy and radiochemotherapy without addition ICI - which have all produced disappointing results. I believe the neoadjuvant trial has to be discussed separately from the trials in advanced PDAC.
2. When discussing T-cell-based immune therapies introduction of the terms antigenicity and immunogenicity is always helpful to explain biological effects of ICI.
Author Response
I appreciate the reviewers for their thoughtful comments about the manuscript.
I have carefully considered the comments and have revised the manuscript according to the reviewers’ comments. The revised portions of the manuscript are in red.
Reviewer 3
This is an informative review paper on a topic that is interesting to clinicians and basic scientists alike.
Major aspects:
- Targeting resectable PDAC in a neoadjuvant setting requires thourough definition of the terms resectable, borderline, advanced. The trial mentioned (Katz et al.) has to be put into context of ongoing trials on neoadjuvant chemotherapy and radiochemotherapy without addition ICI - which have all produced disappointing results. I believe the neoadjuvant trial has to be discussed separately from the trials in advanced PDAC.
Reply: The trial by Katz et al. has been deleted from the Table 1 and discussed separately in the discussion section.
- When discussing T-cell-based immune therapies introduction of the terms antigenicity and immunogenicity is always helpful to explain biological effects of ICI.
Reply: I have revised the relevant sentence as below according to the reviewer’s suggestions.
“Immune-checkpoint inhibitors (ICIs), such as anti-PD-1/PD-L1 and anti-CTLA-4 anti-bodies, have revolutionized cancer treatment, demonstrating durable responses in high immunogenicity tumors such as melanoma and non-small cell lung cancer, and their clinical application is expanding year by year [2]. However, their efficacy in PDAC has been limited due to low antigenicity to trigger an immune response and abundant cancer-associated fibroblasts, both of which result in an immune-suppressive tumor micro-environment (TME) with few tumor infiltrating lymphocytes (TILs)”
Round 2
Reviewer 1 Report
Comments and Suggestions for Authors
I have no further comments.
Author Response
I appreciate your helpful comments to improve my manuscript.
Reviewer 2 Report
Comments and Suggestions for Authors
Line 64-5: What is "spontaneous tumor mouse model derived from pancreatic beta cell"?
Line 104-107:
You wrote "In their experiment, pancreatic cancer cells were first grown in the subcutaneous area of the mouse and then the grown tumors were implanted into the pancreas of another syngeneic mouse. "
This pancreatic cancer cells were primary cells or cell line??
Line 107: X → ×
Line 141: Wouldn't the explanation of the experiment in 3.3 be easier to understand if you used a diagram?
Line170: Why you write "anti-PD-1 antibody" in italic.
Between line 203 and 204, space must be there.
”〜et” al and at the beginning of the sentence is conspicuous. Please attempt to avoid redundancy in your descriptions.
In table 1, "experimentl arm"
Author Response
I appreciate the reviewer 2 for thoughtful comments about the manuscript.
I have carefully read the comments and have revised the manuscript according to the reviewer’s comments. The revised portions of the manuscript are colored in red.
I hope the revised manuscript will now be suitable for publication in Cancers.
Reviewer 2.
Line 64-5: What is "spontaneous tumor mouse model derived from pancreatic beta cell"?
Reply: I have revised the text as follows.
“Klug et al. utilized both a genetically engineered RIP1-Tag5 mouse model mimicking pancreatic insulinoma and a xenografted melanoma tumor to evaluate the effects of low dose RT on tumor-associated macrophages (TAMs).”
Line 104-107:
You wrote "In their experiment, pancreatic cancer cells were first grown in the subcutaneous area of the mouse and then the grown tumors were implanted into the pancreas of another syngeneic mouse. "
This pancreatic cancer cells were primary cells or cell line??
Reply: I have revised the text as follows.
“pancreatic cancer cells of KPC cell line were first grown in the subcutaneous area of the mouse”
Line 107: X → ×
Reply: I have revised it.
Line 141: Wouldn't the explanation of the experiment in 3.3 be easier to understand if you used a diagram?
Reply: I have added a new figure 2 to explain Xie’s trial cohort.
Line170: Why you write "anti-PD-1 antibody" in italic.
Reply: I apologize for my mistake. I have fixed it.
Between line 203 and 204, space must be there.
”〜et” al and at the beginning of the sentence is conspicuous. Please attempt to avoid redundancy in your descriptions.
Reply: I have revised the description of the beginning throughout section 3 to avoid the repetition of the subtitle.
In table 1, "experimentl arm"
Reply: I apologize for my mistake. I have fixed it.
Reviewer 3 Report
Comments and Suggestions for Authors
The revised version is markedly improved, and I support publication.
Author Response

(The authors gave the same response as above.)

Round 3
Reviewer 2 Report
Comments and Suggestions for Authors
Overall, there are too many line breaks, giving the impression of “bullet points”.
There should be a space between Line 203 and 204.
Table 1 would be easier to view if the vertical width were reduced a little, so that the tables are on a single page.
Line 119. Abbreviations are better explained in footnotes than in figures.
What is the underline in line 146, “NCT02311361”?
Line 151, 153: day1 → day-1 (Match the notation on line 150)